# Launching of the Anaemia Research Peruvian Cohort (ARPEC): a multicentre birth cohort project to explore the iron adaptive homeostasis, infant growth and development in three Peruvian regions

Doreen Montag [ID],[1] Carlos A Delgado [ID],[2,3] Consuelo Quispe,[4] David Wareham,[5] Valentina Gallo [ID],[1,6] Jose Sanchez-Choy [ID],[7] Víctor Sánchez [ID],[2] Ruth Anaya,[4] Elaine Flores,[1] Lorena Roca,[4] Víctor Mamani [ID],[8,9] Juan Rivera Medina [ID],[3,10] Pablo Velasquez,[2,11] Carlos Del Aguila [ID],[12,13] Andrew Prendergast,[5] Julio Palomino [ID] [14]

DM and CAD contributed equally.

For numbered affiliations see end of article.

**Correspondence to**
Dr Doreen Montag;
d.montag@qmul.ac.uk

## ABSTRACT

**Background** Preventing infantile anaemia and ensuring optimal growth and development during early childhood, particularly in resource-constrained settings, represent an ongoing public health challenge. Current responses are aligned to treatment-based solutions, instead of determining the roles of its inter-related causes. This project aims to assess and understand the complex interplay of eco-bio-social-political factors that determine infantile anaemia to inform policy, research design and prevention practices.

**Methods** This is a longitudinal birth cohort study including four components: (1) biological, will assess known blood markers of iron homeostasis and anaemia and stool microbiota to identify and genetically analyse the participants' flora; (2) ecological, will assess and map pollutants in air, water and soil and evaluate features of nutrition and perceived food security; (3) social, which will use different qualitative research methodologies to explore key stakeholders and informants' perceptions related to nutritional, environmental and anaemia topics, participant observations and a participatory approach and (4) a political analysis, to identify and assess the impact of policies, guidelines and programmes at all levels for infantile anaemia in the three regions. Finally, we will also explore the role of social determinants and demographic variables longitudinally for all study participants. This project aims to contribute to the evidence of the inter-related causal factors of infantile anaemia, addressing the complexity of influencing factors from diverse methodological angles. We will assess infantile anaemia in three regions of Peru, including newborns and their mothers as participants, from childbirth until their first year of age.

**Ethics and dissemination** Ethical approval was obtained from the Institutional Research Ethics Committee of the Instituto Nacional de Salud del Niño (Lima, Peru), CIEI-043-2019. An additional opinion has been granted by the Ethical Committee of Queen Mary University of London (London, UK). Dissemination across stakeholders is taking part as a continues part of the research process.

---

### Strengths and limitations of this study

► There is scarce empirical evidence on the inter-related causes of infantile anaemia, which will be addressed in this project through an eco-bio-social-political approach.
► Iron supplementation for infants seems not to be a solution in iron deficiency anaemia.
► Prevention measurement effects remain unclear before 6 months of age.
► The project outcomes will inform policy, research design and prevention practices from a multidisciplinary evidence-based perspective.
► The project might be limited by the current COVID-19 pandemic.

---

## INTRODUCTION

Iron deficiency (ID) anaemia (IDA) is a pressing global health challenge, which impacts the auditory, visual and neurological development of children.[1–3] By 2016, the worldwide prevalence of anaemia in children <5 years, reported in the WHO Global Health Observatory data repository was 41.7% (38.1–45.9). The major risk factors for IDA are food insecurity,[4] malnutrition and poverty, all of which will be exacerbated under current and future climate conditions.[5] The anthropocene is marked by global change influencing planetary processes, including biodiversity and land-use changes, impacting food security, soil and water health directly linked to IDA.[6] Characterising and solving the underlying complexity of the factors that interact and determine IDA for millions of children, living in different circumstances globally, will be crucial to adapt to future risks and to

mitigate the burden of disease by developing a durable environmental governance focused on health.[7–9]

In Peru, as in other low/middle-income countries (LMICs), ID and IDA represent a high burden of disease. Between the years 2000 and 2010, the prevalence was over 50% in children aged 6–35 months and by 2017 it remained around 40%.[10] Nationally in the same period, anaemia prevalence decreased by about 10%.[11] However, from 2012 to date, the prevalence rates of anaemia remain around 42.2%.[10 12]

It is important to highlight, though, that treating a deficiency does not necessarily mean supplementing it. During early life, iron homeostasis controls are absent or limited, because the regulation mechanisms for this process develop in later infancy.[13] Prescription of iron supplementation during the first months of life, whether during breast feeding or mixed breast feeding, may interfere with nutrient absorption and may adversely affect the intestinal microbiome.[14 15] Also, there is sparse evidence that iron supplementation may correct neurodevelopmental deficits caused by a lack of iron in early childhood.[16]

### The eco-bio-socio-political framework in Peru
#### Political and economic factors
Over the last decades, there has been a steady increase of the gross domestic product (GDP) in Peru,[17] with up to over 8% per year growth between the years 2000 and 2018[17] reflecting a commodity boom. The GDP, as an indicator of macro-economic development, should have had a direct impact on poverty reduction, decreased food insecurity indicators and improved health outcomes, leading to reduced IDA prevalence. However, this has not happened: high rates of infant anaemia still exist[18] suggesting a lack of a continuous systemic approach by the government, despite its importance on social and economic indicators for the Peruvian society.

#### Social factors
Food insecurity in Peru is rooted in Peruvian colonial history and imbalanced power relations. It is distributed unevenly in each ecological region, often merely addressed as a problem of poverty, which is poorly overseen by policies and interventions over the last 70 years.[19] Food insecurity and malnutrition are consequences of multidimensional factors related to broader social and economic inequalities. These, in turn, influence rural–urban migration patterns and configures budgetary restrictions for food and nutrition of women in their homes.[20 21] Gender relations within households' further influence power dynamics and subsequently has a direct impact on infant nutrition.[22]

#### Biological factors
Biological factors encompass microbiota development and iron homeostasis in breastfed infants. During breast feeding, there is a predominance of *Bifidobacterium* bacteria, that represent 60%–90% of the total microbiota,

due to the rich supply of oligosaccharides in breast milk. As complementary food gets introduced, these are successively displaced by *Firmicutes* and *Bacteroidetes*, capable of absorbing complex polysaccharides derived from plant products appropriate for the stage of infant feeding and growth. Given the large concentration of lactoferrin (an iron-transporting protein) in breast milk, and its resistance to proteolysis, it can be detected in stools of breast-feeding newborns.[23] Optimal microbiota succession can be disrupted by environmental factors, such as exposure to unclean water and lack of sanitation, early introduction of complementary feeding during the period of exclusive breast feeding and the quality of the infant diet.

#### Ecological and environmental factors
Biodiversity loss, increased urbanisation and deteriorating ecosystems have a documented impact on gut microbiota and health.[24 25] Rook and Knight[26] argue that biodiversity of the soil, such as the green spaces that children mature in, plays a vital role in the bacteria-induced immune modulation, and subsequently influences microbiota development and homeostasis. Anthropogenic climate change is altering biodiversity and iron and other mineral content of certain foods, further restricting access to food across vulnerable population groups and increasing food insecurity.[27 28]

Water quality is influenced by biodiversity, deforestation, mining activities and general pollution. Water pollution, specifically, can lead to increased parasitic infections in infants, which in turn can compete for iron in the intestine.[24 29] Environmental enteropathy (EE), also called environmental enteric dysfunction,[30] is a subclinical disorder of the small intestine which alters the structure and function of the gut among children living in settings with environmental contamination and high poverty rates. EE reduced gut barrier function, and intestinal malabsorption and all of these processes may contribute to IDA.[31 32]

#### Rationale of the project
Given the complexity of the factors potentially increasing the risk of IDA, a comprehensive approach is needed to investigate and disentangle such complexity. This entails examining in detail the ecological, biological, social and political factors that would increase the risk for developing IDA in the first year of life among 330 children in Lima (Coast), Ayacucho (Andes) and Pucallpa (Amazon), who will be born to healthy mothers without anaemia during the year 2021. We are proposing an eco-bio-socio-political framework, which is influenced by the work edited by Charron[33] on 'Ecohealth research in practice', that emphasises the importance of participation of all stakeholders and an interdisciplinary methodology, gender equity and sustainability to increase potentially lasting policy impacts. With the base of an initial simple diagram (figure 1) and the research outcomes, we plan to approach and comprehend in-depth the complex

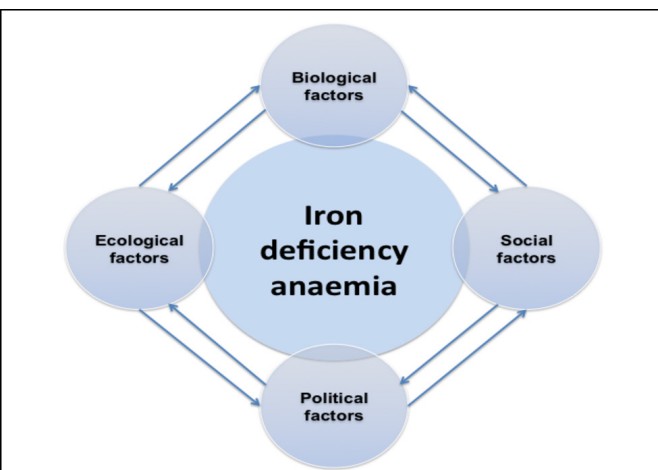

**Figure 1** The eco-bio-socio-political framework applied to iron deficiency anaemia in children in Peru. Source: Own elaboration (DM).

interplay between the selected factors which will allow us to further develop this framework.

The Anaemia Research PEruvian Cohort (ARPEC) project seeks to diminish the scientific and technical evidence-based gap, supporting the development of comprehensive and sustainable policy.[7 20 21 34] In Peru, as in most of Latin America, the majority of anaemia is related to ID, due to lower levels of iron in the diet and its coexistence with high rates of gastrointestinal issues that impairs iron absorption.[35–37] For example, in the Andean region of Ayacucho, children's dietary iron consumption is almost 25% below the recommended iron requirements by age.[38] Despite this, the interdisciplinary empirical literature on the tangled causes and long-term determinants of this highly prevalent challenge is almost non-existent.

The ARPEC project status is updated regularly on its webpage https://www.qmul.ac.uk/arpecproject/.

### Aim and objectives of the study

Understand the complexity of the eco-bio-socio-political factors that influence food insecurity of breastfeeding mothers and their newborns in three Peruvian regions, altering mechanisms of adaptive iron homeostasis in infants within the anthropocene.

### Objectives

1. To evaluate the effects of infant exposure to ecological/environmental factors, including nutrition, water quality and air and soil pollution, and their potential influence on gut microbiome development and the mechanisms of adaptive iron homeostasis in infants.
2. To assess the contribution of biological factors, such as succession of the intestinal microbiota, to the incidence of IDA in breastfed infants during the first year of life. We will focus especially on those organisms with siderophore genes and a competitive role in intestinal iron absorption.
3. To identify social factors influencing dietary patterns and food insecurity and their association with ID or IDA in breastfed babies during the first year of life.
4. To analyse political factors, including current policies and success or failure of previous policies and lasting and sustainable policies, in addressing food insecurity and IDA in different Peruvian regions.
5. To develop a framework able to explain the complex interaction and interdependence of the eco-bio-social-political factors determining IDA in Peru.

## METHODS

### Study design

ARPEC project is a prospective observational multicentre birth cohort study that will be conducted in three regions within Peru: the coast (Lima), the Andes (Ayacucho) and the Amazon rainforest (Pucallpa) with integrated quantitative and qualitative methodologies. The recruitment of participants for the birth cohort study is expected to start in July 2021 and overall data selection to be finished by August 2022. Literature review, policy analysis and qualitative interviews with stakeholders have started earlier and are expected to be culminated after.

### Setting

Peru is the third most megadiverse country in the world, home to 55 indigenous groups (51 in the Amazon and 4 in the Andes) with 48 different indigenous languages.[39] Diversity is reflected in the nutritional and food practices throughout the country. This study involves inhabitants from three different regions: (1) Lima, as the capital of Peru, is located at 101 m above sea level in the central coast. Lima encompasses almost 9.5 million inhabitants (32.3% of the national population) in a desert environment. It has the lowest proportion nationally of children <5 years old (7.3%).[40] By 2018, in the urban Lima, the prevalence rates of anaemia in children between 6 and 35 months was 35.9%[12] and in 2017, among women between 15 and 49 years old was 23.1%.[10] Stunting for Lima in 2019 was reported as 4.9%.[12] (2) Pucallpa, the capital of the regional department of Ucayali, located at 154 m above sea level, in the central Peruvian Amazon rainforest. It has almost half a million inhabitants (1.7% of the national population). The proportion of children <5 years old in Ucayali is 11.8%.[40] Stunting among children <5 was 24.8%, one of the highest in the country. The prevalence of anaemia in children between 6 and 35 months was 56.4% in 2018,[12] and the highest prevalence was observed in children between 0 and 8 months (62%).[11] (3) Ayacucho, is located at 2760 m above sea level in the Andes of South-eastern Peru. This region was the most affected during the bloody internal conflict in Peruvian recent history (1980–2000). The region has 616176 inhabitants (2.1% of the national population). The proportion of children <5 years old in Ayacucho is 8.8%.[40] In 2018, stunting prevalence was 18.5% in children <5 years of age, and anaemia affected 49.3% of

children aged 6–35 months.[12] According to the 2017 Census (INEI 2017) Ayacucho women of childbearing age constitute almost 25.5% of the total population (in contrast to 26.7% nationwide). Almost 27% of women from Ayacucho were mothers.[41]

## Participant recruitment and selection criteria

The project will identify, select and recruit 330 pairs of mothers and newborns who will be involved in the study. Potentially eligible participants will be identified from the official registries of full-term pregnancies at the recruitment centres in the three cities. The recruitment centres are second level health establishments that organise the identification and monitoring of prenatal controls for pregnant women in their area of influence. Pregnancies with risk factors are referred to facilities with greater resolution capacity. We will consider the following inclusion criteria: singleton births, delivered vaginally, with umbilical cord clamping after ≥60 s, newborns weighing >2500 g, mother–baby hospitalised jointly and discharged before 72 hours of life, without diagnosis of perinatal infection, birth asphyxia or congenital malformation. Participant mothers aged 18+ years, who are capable of understanding study procedures, providing informed consent and full-time residents in the study area. The mothers will be assessed during the month before the childbirth, at the last prenatal check-up or retrospectively through review of clinical records. Exclusion criteria are: mothers with a diagnosis of anaemia on the last prenatal haemoglobin (Hb) levels; who had delivery via caesarean section; received antibiotics during the last month of pregnancy and the first week of life (mothers or their newborns). The selection criteria are based on a low level of complexity for primary or secondary level of care. Certainly, at a hospital in Lima there is a greater number of patients who end up in caesarean section (because they tend to be more difficult cases), but even in a national reference hospital, there are also enough cases of low level of complexity. It is expected that recruitment participants from a similar level of complexity might minimise the selection bias.

## Sample size and power calculation

The study sampling will be purposeful. We plan to have a total sample of 330 mother/child pairs, recruiting 110 newborns from each region, totalling 660 participants and assuming a maximum potential loss of follow-up of 10%. Our hypothesis is that selected ecological, biological, social and political factors increase the risk of IDA in the first year of life among the newborn's participants from the three regions. To test this association, we will have a power of 90% at a significance level of 5%, to detect an average difference of 0.5 g/L between two subgroups, and 75% power to detect an average difference of 0.4 g/L. Both are important differences for public health decisions and are associated with relevant outcomes such as early childhood development. We estimate that the level of Hb in infants will be similar to that of their mothers, as has been reported for the city of Pucallpa in Peru[42] (average 11.5 g/L, Standard Deviation (DS) 1.3). The sampling frame does not attempt to produce a sample which is proportional to the underlying population, rather a sample which maximises power to detect differences across regions (although not powered to formally test for interaction). Samples of similar size across the recruiting centres will maximise the ability of describing existing differences in one or more variables.

## Patient and public involvement

The research question and methodology were developed and discussed by the principal investigators and co-investigators across the participating four institutions in Peru and the UK. The study does not involve patients. A discussion of the methodology with stakeholders at the beginning of the research project has informed the methodology and is informing the iterative process of the research process.

## Study procedures

### Current knowledge review

We will perform an in-depth review of the literature, combining a systematic search and review with a policy analysis. The systematic review will focus on current knowledge on ID and IDA in Peru and the impact of environmental factors on IDA. The policy analysis will be focused on stakeholders, content, context and capacity analysis to understand past successes and the underlying reasons on why, where and what has not been successful tackling IDA in Peru.[43 44] Findings from reviews will inform the qualitative research component planning and methodology, comprising in-depth interviews (IDIs) and focus groups discussion (FGDs) sessions.

### Assessments of anaemia

We will assess IDA through Hb, ferritin and C reactive protein (CRP) levels in blood samples at baseline (birth) and during follow-up study visits at 4, 8 and 12 months of age.[45] Mothers will be examined after childbirth and before hospital discharge. In addition, concentrations of serum hepcidin, soluble transferrin receptor and lead will also be evaluated. We will use WHO criteria to adjust ferritin values for inflammation,[46] as reflected by the CRP.[47] We will follow standard guidelines to draw laboratory samples from children, accompanied by their mothers or primary caregivers.

### Basic anthropometric measurements

We will measure weight and length of children at baseline and at each follow-up visit, with a digital calibrated scale and an infant metre, respectively, following standard local procedures. Additionally, we will measure head circumference and abdominal circumference with a non-elastic metric tape. Weight, height and abdominal circumference of mothers will be taken at baseline and at the 12-month visit, following standard assessment procedures for adults. All measurements will be performed by trained fieldworkers, with experience in local health services.

Child development will be assessed using a locally validated version of the Child Welfare Survey (SWYC).[48] The SWYC is a free-screening instrument and requires almost 15 min to complete. It includes questionnaires adapted to age milestones to assess three domains of child functioning: developmental domain, emotional/behavioural domain and family context.

## Metagenomic analysis of the microbiota

The child's stool microbiota will be analysed using whole genome sequencing techniques during the three follow-up visits at 4, 8 and 12 months of life. The samples will be collected, processed, stored and shipped for analysis from Peru to the Blizard Institute, Queen Mary University of London (QMUL). We will also undertake phenotypic analysis, using selective culture media, to isolate viable bacterial pathogens in a central laboratory in Lima. Selective chromogenic bacterial culture media will be used for primary isolation and quantification of *Enterobacterales*, *Enterococci*, *Bifidobacterium*, *Lactobacilli*, *Bacteroides*, *Clostridia*, *Candida* spp and siderophore-producing isolates (CAS agar). Molecular analysis and next generation sequencing of the faecal microbiome will be performed by metagenomic shotgun sequencing.[49 50] The DNA of the entire bacterial community will be extracted, fragmented and sequenced using Illumina HiSeq technology. Bioinformatic analysis of sequence data will be performed using the CosmosID Metagenomics Cloud application (https://www.cosmosid.com/shotgun-bioinformatics) (Rockville, Maryland, USA) to enable taxonomic profiling of bacteria, viruses, phages, fungi, protists, microbial resistome and virulome.

## Water quality analysis

Sampling for water quality analysis will be conducted in two different seasons (dry and wet) and within comparative geographical reference points in each region (Pucallpa, Ayacucho, Lima), considering their relationship to other human activities, including mining, lack of sewage and other sanitary installations that have an impact on water quality. We will determine during fieldwork collection the pH, conductivity, turbidity and percentage of dissolved oxygen in water samples. In addition, we will assess under laboratory conditions the levels of metals (such as iron, manganese, aluminium, copper, zinc, lead, mercury), hardness characteristics, presence of nutrients (such as phosphate, nitrate) and microbiologic characteristics (such as the presence and quantity of heterotrophic bacteria, total coliforms and thermotolerant coliforms) following the government's permitted parameters.[51] The assessments will follow the protocols of Standard Methods for Water and Wastewater[52] and laboratory guidelines of the WHO.

## Air quality analysis

We will evaluate air quality across the three regions using data from different monitors, satellite reports and field collection at pre-identified geographical landmarks during the study visits. Environmental exposure to air pollutants, fine particles equal to or less than 2.5 µm in diameter (PM2.5) and $NO_2$ may produce systemic inflammation and determine changes in the bone marrow and act as a direct risk factor for the development of anaemia.[53 54] A recent study conducted in Lima, Peru, determined a significant association between the increase in the level of PM2.5 measured in the environment, with decreased Hb values and an increase in the prevalence of moderate and severe anaemia in children under 5 years.[55]

## Geographic mapping

We will also explore the proposed relationship between environmental exposures and microbiota characteristics using the Geographic Information System Mapping (GIS) for georeferencing. This will allow us to map the prevalence of anaemia through estimating the approximate distance to the child's home, the specific microbiota differences and the biodiversity level of the green areas surrounding the households.[26]

## Sociodemographic and nutritional evaluation procedures

We will assess socioeconomic and food practices variables among all participants. The nutritional instrument will be locally piloted and adapted for each region to provide information on the food groups consumed by the mother and infant beyond 4 months of age when new food is being introduced. This will vary within regions, by seasonality factors and food access. We will also use locally developed visual aids materials to ease the description and characteristics of foods, drinks and portion sizes of the food commonly consumed by the participants.

We acknowledge the potential importance of the participant's various sociocultural backgrounds (Shipibo-Konibo, Quechua) and the different feeding practices during weaning, which will be explored in-depth through qualitative methodology. Our approach aims to be participatory since inception, following Eco health approaches[33] and has a high potential to develop participatory and lasting prevention policies. We will base our analysis on the known nutritional content of fruits and legumes nationally.[56]

## Qualitative research procedures

We will conduct IDIs with participating mothers, regional and national key informants and local stakeholders. We will aim to interview between 25 and 30 mothers (8–10 from each region) to reach saturation, selected by convenience. For the stakeholder interviews, one key stakeholder per institution will be selected. Given the high fluctuance of stakeholders in institutions and to comprehend the challenges of anaemia policies from a historic perspective, a snowball method will be used to determine stakeholders and provide a comprehensive insight.[57] The purpose of interviews among stakeholders is to evaluate their perceptions of what are the causes for IDA and their opinion on the historical success or failure of policy interventions related to anaemia. Three to six FGDs[58] will be

conducted among mothers and stakeholders, separately, in each location to identify, discover and review important themes and provide meaning.[59] FGDs among key stakeholders in each region and on a national level will address two main areas: first, the perceived relationship between nutritional and environmental factors and the potential risks for the development or worsening of the IDA among children. Second, they will address the policy level, which will be divided into two phases: (1) the experiences and perceptions of effective past and current policies and barriers to effective implementation of governmental interventions and lasting policies and (2) during the last part of the research period, after the presentation of the results, to discuss and inform potential policy implementation based on the research results and practical aspects of transforming policies into practice. Finally, we anticipate conducting a public engagement and participatory activity with a group of participating mothers from each region. This will use photography as a means of non-verbal expression, allowing participants to identify, select, register and present everything that they consider important and related to the health of their newborns and the risks related to the development of anaemia, following the PhotoVoice method.[60]

### Operationalisation of the variables
The list of variables and their characteristics will be included in the study protocol, instruments, manuals and standard operations procedures.

### Data analysis
We will conduct biological (microbiota/stool, water, blood analysis), sociocultural (nutrition, environment, policy), geographic mapping (GIS–location relationship between green spaces, biodiversity and gut microbiota) and mixed-methods research (nutrition/household survey, qualitative methodology).

### Quantitative data analysis
We will review the distribution of any missing values and evaluating imputation options, normality for all quantitative variables and their correlations. We will conduct descriptive analyses of the distribution of all microbiome and lifestyle data between the different Hb levels across all participants.

Surveyed quantitative variables will be categorised appropriately for each follow-up visit and we will estimate their changes over time. Biological samples will be analysed to evaluate its microbiota content, and we will identify the most frequent taxa for each sample. Repeated measurements will allow us to calculate the largest changes in microbiota composition during follow-up period. Principal component analysis will be used to identify the most distinctive taxa and their associated patterns of changes with the lowest Hb levels at the end of the study period. These in turn will be analysed in association with other lifestyle variables to identify possible causal pathways.

A multivariate logistic regression model will be used to investigate the cross-sectional association between potential risk factors and Hb levels in children, in each visit. Additionally, information on selected exposures will be collected four times to calculate cumulative exposure, which will be then evaluated against ID and IDA at the end of study period. Both cumulative exposures and their changes will be included in a final Cox model, including exposure variables at the starting point, and their relevant changes, to explore the risk associated with lower Hb levels, during the first year of life.

### Qualitative analysis
Policy analysis will be conducted, encompassing national and regional policies since 1979 to comprehend the development of pData obtained from participant observations, FGD sessions and IDIs will be audio-recorded and transcribed verbatim. This will later be segmented, entered, coded and analysed to identify a list of relevant themes and key concepts related to the topics of interest. Coding will continue throughout the research process, using the grounded theory approach to data analysis, and informing follow-up interviews and focus groups.[61] The analysis will compare the information by different subgroups of participants to identify and describe the similarities and divergences between key stakeholders, participants from different regions, age groups and those living across different environmental and nutritional

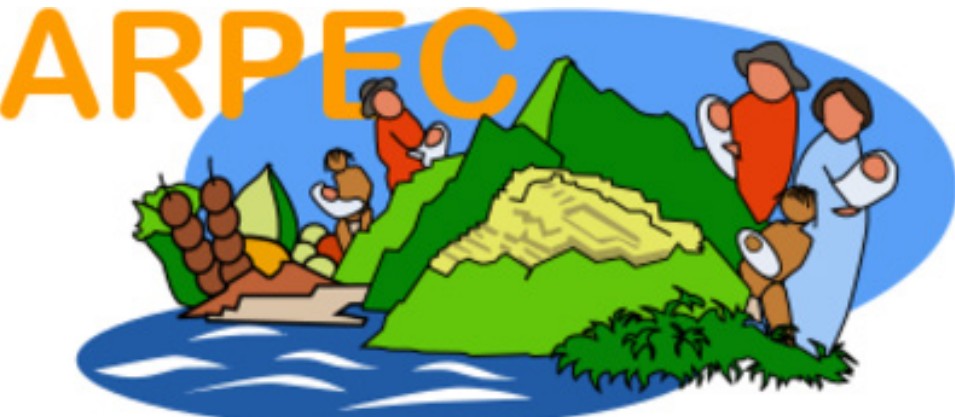

**Figure 2**  ARPEC logo by Daddo illustrations at www.daddoillustrations.com.

risks. The analysis will also include field notes, direct observation and minutes of the researchers' meetings.

## ETHICS

Ethical approval was obtained from the Institutional Research Ethics Committee of the Instituto Nacional de Salud del Niño (Lima, Peru), CIEI-043-2019. A second opinion has also been granted by the Ethical Committee of Queen Mary University of London (London, UK). Further to that, institutional support was sought from local authorities, particularly indigenous government organisations, local hospitals and health centres.

Written informed consent will be sought from all participants prior to enrolment. Verbal consent and permanent explanation of research purposes will continue throughout the study.

To protect participant's confidentiality, all collected data will be linked to a unique identification code. Data will be recorded in paper form and electronically. The electronic information will be stored, backed-up and secured by password protection in the RedCap server. Paper forms will be stored in secure locked cabinets at the three Peruvian sites. All confidential information, including participant's contact details will also be stored in secured locations, accessible only to authorised research staff. Long-term data management will comply QMUL's research policy.

Any requests for biological samples will be processed following international standards. Importantly, only processed stool samples will be sent to the UK.

### Plan for results dissemination

The study team has been following and will adhere to a detailed plan for dissemination of study outcomes, that will be used to publicise the research at a national and regional level. The outline dissemination methods will include periodic meetings with stakeholders, academic institutions that directly and indirectly collaborate with the study team, scientific publications in peer-reviewed journals, presentation in scientific events to allow engagement with the wider international academic society in the discussion of research results, creation and release of integral policy briefs for stakeholders at the regional and national level in Peru, teaching through research-based teachings in relevant participating institutions and modules and presence in social media and a dedicated study website and logo (figure 2).

**Author affiliations**
[1]Centre for Global Public Health, Queen Mary University of London, London, UK
[2]Department of Medicine, Neonatal Intensive Care Unit, Instituto Nacional de Salud del Niño, Lima, Peru
[3]Department of Paediatrics, Faculty of Medicine, Universidad Nacional Mayor de San Marcos, Lima, Peru
[4]School of Nursing, Universidad Nacional de San Cristóbal de Huamanga, Ayacucho, Peru
[5]Blizard Institute, Queen Mary University of London, London, UK
[6]Campus Fryslan, University of Groningen, Leeuwarden, The Netherlands
[7]Department of Aquaculture and Agroforestry, Universidad Nacional Intercultural de la Amazonia, Pucallpa, Peru
[8]Executive Office for Research Support and Specialized Teaching, Instituto Nacional de Salud del Niño, Lima, Peru
[9]School of Nutrition and Dietetics, Universidad Científica del Sur, Lima, Peru
[10]Department of Medicine, Gastroenterology, Hepatology and Nutrition Unit, Instituto Nacional de Salud del Niño, Lima, Peru
[11]Department of Neonatal Medicine, Instituto Nacional Materno Perinatal, Lima, Peru
[12]Department of Medicine, Endocrinology and Metabolism Unit, Instituto Nacional de Salud del Niño, Lima, Peru
[13]Faculty of Medicine, Universidad Nacional Federico Villarreal, Lima, Peru
[14]Faculty of Environmental Sciences, Universidad Nacional Santiago Antúnez de Mayolo, Huaraz, Peru

**Acknowledgements** We would like to thank stakeholders for their constructive feedback on our methodology and taking part in the iterative participatory process.

**Contributors** DM conceived study ideas, drafted protocol and overall responsibility of study design and implementation. Drafted the current manuscript. CD conceived study ideas, major contribution to protocol and overall responsibility of study design and implementation in Peru. CQ is responsible for conceiving and operationalising anthropometric data collection. DW is responsible for conceiving and operationalising genetic analyses of the microbiome. Theoretical and logistic input on faeces sample collection and microbiota analyses. JS-C is a manager of data collection in Pucallpa. VS contributed to overall input on study design and biomedical background and measurements. RA contributed to general input on operationalising anthropometric data collection. VG contributed to overall advice on epidemiological study design and statistical analyses. General input on biomedical background and measurements for neurodevelopmental issues. AP contributed to overall input on study design and implementation. LR contributed to general input on anthropometric data collection and study context in Ayacucho. EF is responsible for coordination of protocol writing, collaborating in this manuscript drafting. VM is responsible for nutrition expert, intellectual contribution to nutritional component. JRM is responsible for intellectual contribution to biological research component. PV is responsible for intellectual contribution to biological research component. CDA is responsible for intellectual contribution to biological research component. JP is responsible for conceiving and operationalising environmental data collection. Theoretical and logistic input on faeces sample collection, microbiota and environmental analyses.

**Funding** This project is supported by the Medical Research Council—MRC grant number MR/S024654/1 and by the Fondo Nacional de Desarrollo Científico, Tecnológico y de Innovación Tecnológica—FONDECYT, grant number 033-2019. AJP is funded by the Wellcome Trust (108065/Z/15/Z).

**Competing interests** None declared.

**Patient and public involvement** Patients and/or the public were not involved in the design, or conduct, or reporting, or dissemination plans of this research.

**Patient consent for publication** Not required.

**Provenance and peer review** Not commissioned; externally peer reviewed.

**ORCID iDs**
Doreen Montag http://orcid.org/0000-0003-1365-1913
Carlos A Delgado http://orcid.org/0000-0002-6073-8109
Valentina Gallo http://orcid.org/0000-0002-1268-8629
Jose Sanchez-Choy http://orcid.org/0000-0003-3376-590X
Víctor Sánchez http://orcid.org/0000-0002-0965-310X
Víctor Mamani http://orcid.org/0000-0002-5508-0883
Juan Rivera Medina http://orcid.org/0000-0003-3562-7914
Carlos Del Aguila http://orcid.org/0000-0002-5345-5995
Julio Palomino http://orcid.org/0000-0002-4589-6774

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
