## [Reviewer comments · BMJ Open]

ARTICLE DETAILS

TITLE (PROVISIONAL)	Launching of the Anaemia Research PERuvian Cohort (ARPEC): A multicentre birth cohort project to explore the iron adaptive homeostasis, infant growth, and development in three Peruvian regions.
AUTHORS	Montag, Doreen; Delgado, Carlos; Quispe, Consuelo; Wareham, David; Gallo, Valentina; Sanchez-Choy, Jose; Sanchez, Victor; Anaya, Ruth; Flores, Elaine; Roca, Lorena; Mamani, Victor; Rivera Medina, Juan; Velasquez, Pablo; Del Aguila, Carlos; Prendergast, Andrew; Palomino, Julio

VERSION 1 – REVIEW

REVIEWER	Ann Von Holle National Institute of Environmental Health Sciences, United States
REVIEW RETURNED	27-Nov-2020

GENERAL COMMENTS	This protocol article describes a study with a thorough array of measures to assess iron deficiency in a Peruvian cohort. The well-written article was a pleasure to read. The combination of both qualitative and quantitative measures is innovative and once complete, will go far in advancing an understanding of infantile anaemia to promote better policy, research practices and prevention. I have some questions and/or minor points about the manuscript proposed measures in the protocol as well as the potential for others that were not included. Abstract line 29 I don't understand the word 'natural' in this context. What regions in Peru are not natural? Summary lines 48-52 There are two strengths, which I believe are certainly justified, but why are no limitations listed? What about generalizability to other geographic areas, including inside and outside of Peru? Introduction General comment re the listed domains Why is there no mention of areas related to genetic or epigenetic factors relating to iron homeostasis? For example, HFE mutations, although rare, are related to iron homeostasis. Even if the sample size is not sufficient for genetic analyses in this sample, these data could be eligible and an important contribution to consortium work? lines 13-15 Although I understand this sentence, I am not sure how this
--

	sentence with 'Anthropocene' relates to IDA. Would be nice to reword in more accessible language. Lines 91-32 I appreciate the mention of supplementation not necessarily being a solution. When I read this paragraph, I wondered are the prevention measures if supplementation is not warranted. If prevention strategies remain unclear, then this could tie into the motivation for this study. Political and economic factors, lines 100-106 I am unsure about the assumption in this paragraph that higher GDP was have a direct impact on poverty reduction, etc... First, if income inequality is high and wealth remains concentrated in a small number of people I don't see the economic situation of people with lower socioeconomic position improving. Also, with economic development, would there be a stronger tendency to adopt a Western diet and lifestyle linked with worse health outcomes. Economic and social factors, lines 108-114 These paragraphs are examples of strengths present in this protocol with a comprehensive assessment of factors relating to health, including socioeconomic position, food insecurity, and environmental contamination. Lines 150-151 Figure 1 represents the interplay between factors and IDA, but I am having trouble interpreting this figure because there are no arrows leading to or from the IDA object. I do see the four groups of factors overlapping with IDA, but it would be nice to have some arrows also pointing towards IDA as an outcome, similar to a directed acyclic diagram. Aim and objective of the study Objectives, lines 179-180 If this is a prospective study, its' not clear to me how it will be possible to interpret the effect of previous political factors on IDA without considering what is currently occurring, which is not mentioned in this objective. Methods Setting, lines 205-206 This line seems out of place in the context of the preceding sentence. Participant recruitment and selection criteria, line 211 I'd like to have a sentence or two with a fuller description of the recruitment centres, considering they will be crucial to collection of many measures. Assessment of anemia, lines 246 The inclusion criteria include hospitalized mother-infant pairs. What differences in hospitalization occur across the three regions? I am wondering if proportionately more women are excluded from study dependent on region, and the implications on selection bias present in the study. Participant recruitment and selection criteria, lines 213 The inclusion criteria include hospitalized mother-infant pairs. What
--	---

	differences in hospitalization occur across the three regions? I am wondering if proportionately more women are excluded from study dependent on region, and the implications on selection bias present in the study. Participant recruitment and selection criteria, lines 217-220 I get the impression that formula feeding women are excluded from study given the aims only mention breastfed infants. More detail explicitly stating formula feeding is an exclusion criteria would be helpful in better understanding. Also, the proportion of mother who breastfeed by region would be helpful in the setting description on lines 188-206. Sample size and power calculation, line 223-224 This sampling design appears to oversample certain regions, given the size of Lima relative to the other two regions. More information on the rationale behind this choice would be helpful. Sample size and power calculation, line 227-229 I'm assuming the power calculations for entire sample, but if you stratify by the three regions, what would be power be in that scenario? I do not have enough information to understand why pooling across these three regions is justified, considering their differences relative to potential confounders and the degree to which they would reflect relevant target populations. Metagenomic analysis of the microbiota, lines 263-264 Where are these visits occurring? Geographic mapping, lines 301-302 Prevalence after 'anaemia' appears to be a typo. Sociodemographic and nutritional evaluation procedures, line 310 'locally plot visual'. These words do not make sense to me. Quantitative data analysis, line 353 'We will conduct descriptive analyses' instead of 'We will conduct descriptive analysis'? Quantitative data analysis, line 366 If you have time-dependent covariates I would change 'Cox proportional hazards model' to 'Cox model', since the PH assumption would no longer hold in that case.
--	---

REVIEWER	Heike Rabe Brighton and Sussex Medical School, UK
REVIEW RETURNED	04-Dec-2020

GENERAL COMMENTS	This is a very well and clearly written manuscript about a prospective birth cohort study on influencing factors of anemia in infancy. I have only one minor comment: In the description of the population in region 1, there might be a printing error? proportion of infants > 5 years, should be < 5 years? Do the authors know the proportion of stunting for region 1, as they have described for region 2 and 3? Could they also provide the proportion of infants < 5 years for region 2 and 3? Then the description of the recruitment populations from the 3 regions would be cohesive. If any of this data is not known that could be stated in the description.
---

VERSION 1 – AUTHOR RESPONSE

Reviewer: 1

Comments to the Author:

Abstract

line 29

I don't understand the word 'natural' in this context. What regions in Peru are not natural?

*The denomination of natural region refers to units of territory (Campos 2018) that have common geographic or climatic characteristics. It does not imply the absence of nature in that area. However, to avoid confusion of terms, we have excluded the “**natural**” word from the abstract.*

(Campos 2018) <https://doi.org/10.1515/geo-2018-0004>

3. Summary

lines 48-52

There are two strengths, which I believe are certainly justified, but why are no limitations listed? What about generalizability to other geographic areas, including inside and outside of Peru?

We have now included the following:

Limitations: *The first great limitation to field research in the 2020s is the COVID-19 pandemic. Recruitment in the hospitals and visits during follow-up must be performed under extreme safety precaution against covid transmission. In a recent steering meeting, the argument of restricting our study population to carry it out was reinforced. The restriction may imply conducting the study just in two cities instead of three. If conditions continue to be adverse, it will also include a reduction in the sample size and procedures to be performed. However, in February 2021 the health personnel started vaccination in Peru and many of our researcher staff are related to health facilities. Consequently, we still hope to start recruiting in July 2021. The disparities among participating centres' laboratory facilities will also generate additional challenges to pick and ship the biological samples.*

Generalizability: *To obtain results replicable over space and time, it is mandatory to include any influential variant to achieve sufficient explanation to be true. The study uses an approach to comprehend the complex interplay of eco-bio-social-political factors on anemia in particular settings. While the approach as a methodology might show very useful to adapt to diverse settings the interplay of the factors is likely going to be different in diverse settings and might need additional studies and might therefore not be generalizable.*

4. Introduction

General comment re the listed domains

Why is there no mention of areas related to genetic or epigenetic factors relating to iron homeostasis? For example, HFE mutations, although rare, are related to iron homeostasis. Even if the sample size

is not sufficient for genetic analyses in this sample, these data could be eligible and an important contribution to consortium work?

Hemochromatosis or HFE mutations are extremely rare, and a small sample size probably will not include extreme values in ferritin or similar exams to guide this type of diagnosis. Additionally, a pilot study like ours may not include specific criteria for a consortium work analysis.

5. lines 13-15

Although I understand this sentence, I am not sure how this sentence with 'Anthropocene' relates to IDA. Would be nice to reword in more accessible language.

The sentence has been modified to: "The Anthropocene is marked by global change influencing planetary processes, including biodiversity and land-use changes, impacting food security, soil and water health directly linked to IDA."

6. Lines 91-32

I appreciate the mention of supplementation not necessarily being a solution. When I read this paragraph, I wondered are the prevention measures if supplementation is not warranted. If prevention strategies remain unclear, then this could tie into the motivation for this study.

We agree with this statement as the main research proposal for our project. Iron supplementation for infants seems not to be a solution in iron-deficiency anaemia, and prevention measurement effects remain unclear before six months of age. This has now been included in the summary.

7. Political and economic factors, lines 100-106

I am unsure about the assumption in this paragraph that higher GDP was have a direct impact on poverty reduction, etc... First, if income inequality is high and wealth remains concentrated in a small number of people I don't see the economic situation of people with lower socioeconomic position improving. Also, with economic development, would there be a stronger tendency to adopt a Western diet and lifestyle linked with worse health outcomes.

We agree with this, which is actually the main point, that besides the GDP increase there has not been any decrease of inequality or poverty or food insecurity. This is exactly what is written in the paragraph "Over the last decades there has been a steady increase of the Gross domestic product (GDP) in Peru¹⁷, with up to over 8% per year growth between the years 2000 and 2018¹⁷ reflecting a commodity boom. The GDP, as an indicator of macro-economic development, should have had a direct impact on poverty reduction, decreased food insecurity indicators and improved health outcomes, leading to reduced IDA prevalence. However, this has not happened: high rates of infant anaemia still exist¹⁸ suggesting a lack of a continuous systemic approach by the government, despite its importance on social and economic indicators for the Peruvian society. „

8. Economic and social factors, lines 108-114

These paragraphs are examples of strengths present in this protocol with a comprehensive assessment of factors relating to health, including socioeconomic position, food insecurity, and

environmental contamination.

We agree. No additional comments.

9. Lines 150-151

Figure 1 represents the interplay between factors and IDA, but I am having trouble interpreting this figure because there are no arrows leading to or from the IDA object. I do see the four groups of factors overlapping with IDA, but it would be nice to have some arrows also pointing towards IDA as an outcome, similar to a directed acyclic diagram.

In the framework, IDA is currently overlapping showing the basic interrelationship of each factor with IDA. It is by purpose that the framework is not more detailed and does not contain any further arrows as this is the data that we will add based on our research results, we have therefore not changed the framework.

Aim and objective of the study

10. Objectives, lines 179-180

If this is a prospective study, its' not clear to me how it will be possible to interpret the effect of previous political factors on IDA without considering what is currently occurring, which is not mentioned in this objective.

The objective aims at taking a historical perspective on current context and policies, which includes a historical and current analysis of political factors. We have tried to make this clearer now: "To analyse political factors, including current policies and success or failure of previous policies and lasting and sustainable policies, in addressing food insecurity and IDA in different Peruvian regions."

Methods

11. Setting, lines 205-206

This line seems out of place in the context of the preceding sentence.

IT SAYS: Ayacucho women of childbearing age constitute 17% of the total population (21% nationwide) and 31.5% of pregnant women.

It is now changed to: According to the 2017 Census (INEI 2017) Ayacucho women of childbearing age constitute almost 25.5% of the total population (in contrast to 26.7% nationwide). Almost 27% of women from Ayacucho were mothers.

INEI. Censos Nacionales 2017 URL: < <http://censo2017.inei.gob.pe/> >

12. Participant recruitment and selection criteria, line 211

I'd like to have a sentence or two with a fuller description of the recruitment centres, considering they will be crucial to collection of many measures.

We have now included the following: "The recruitment centers are second level health establishments that organize the identification and monitoring of prenatal controls for pregnant women in their area of influence. Pregnancies with risk factors are referred to facilities with greater resolution capacity."

13. Assessment of anemia, lines 246

The inclusion criteria include hospitalized mother-infant pairs. What differences in hospitalization occur across the three regions? I am wondering if proportionately more women are excluded from study dependent on region, and the implications on selection bias present in the study.

The comment under 13 and 14 are the same so we are addressing them under point 14 for both.

14 Participant recruitment and selection criteria, lines 213

The inclusion criteria include hospitalized mother-infant pairs. What differences in hospitalization occur across the three regions? I am wondering if proportionately more women are excluded from study dependent on region, and the implications on selection bias present in the study.

We have now included a more detailed description:

"The selection criteria are based on a low level of complexity for primary or secondary level of care. Certainly, at hospital in Lima there is a greater number of patients who end up in cesarean section (because they are more difficult cases), but even in a national reference hospital, there are also enough cases of low level of complexity. It is expected that recruitment participants from a similar level of complexity might minimize the selection bias."

15. Participant recruitment and selection criteria, lines 217-220

I get the impression that formula feeding women are excluded from study given the aims only mention breastfed infants. More detail explicitly stating formula feeding is an exclusion criteria would be helpful in better understanding. Also, the proportion of mother who breastfeed by region would be helpful in the setting description on lines 188-206.

We did not consider the formula feed infants as criteria for exclusion, because we would like to assess results with minimal intervention. We also, will register every type of milk received by participants in order to assess its short term effect.

According to the 2019 Demographic and Health Survey (INEI 2019), the estimated value of the proportion of exclusive breastfeeding at 6 months (CI 95%) for Lima, Ayacucho and Pucallpa was 54.7% (CI95= 46.6; 62.8), 73.5% (CI95= 61.5; 85.4) and 74.4% (CI95= 66.8; 82.0), respectively.

16. Sample size and power calculation, line 223-224

This sampling design appears to oversample certain regions, given the size of Lima relative to the other two regions. More information on the rationale behind this choice would be helpful.

We have now included the following: *“The sampling frame does not attempt to produce a sample which is proportional to the underlying population, rather a sample which maximises power to detect differences across regions (although not powered to formally test for interaction). Samples of similar size across the recruiting centres will maximise the ability of describing existing differences in one or more variables.”*

17. Sample size and power calculation, line 227-229

I'm assuming the power calculations for entire sample, but if you stratify by the three regions, what would be power be in that scenario? I do not have enough information to understand why pooling across these three regions is justified, considering their differences relative to potential confounders and the degree to which they would reflect relevant target populations.

Given the complexity of the framework, and the number of biological samples and follow-up visits required to potentially disentangle it, the overall sample size could not be larger. However, it would allow enough power to detect main risk factors associated to IDA at country level, although it won't be enough to test for interaction. However, the richness of the data collected (i.e. the microbiome of the soil and the qualitative analysis of cultural believes in relation to IDA) will also allow generating interesting hypothesis on further local factors associated with IDA, beyond the nation-wide overall model. These however, will probably required to be tested in separate studies. Under current circumstances marked by the pandemics, even an exploratory pilot cohort study like ours may be useful to generate new explanations or hypotheses to be tested in further anaemia research.

18. Metagenomic analysis of the microbiota, lines 263-264

Where are these visits occurring?

Visits will occur every four months until one year of age, at home and at the health facility, conducting interviews, anthropometry, taking samples and other actions according to what is programmed at all times.

19. Geographic mapping, lines 301-302

Prevalence after 'anaemia' appears to be a typo.

Yes it is a typo.

It is changed to: ... map the prevalence of anaemia ~~prevalence~~ ...

20. Sociodemographic and nutritional evaluation procedures, line 310

'locally plot visual'. These words do not make sense to me.

It is changed to: ... locally developed visual aids ...

21. Quantitative data analysis, line 353

'We will conduct descriptive analyses' instead of 'We will conduct descriptive analysis'?

It is changed to: 'We will conduct descriptive analyses'

22. Quantitative data analysis, line 366

If you have time-dependent covariates I would change 'Cox proportional hazards model' to 'Cox model', since the PH assumption would no longer hold in that case.

We agree, and that has now been changed.

Reviewer: 2

1. In the description of the population in region 1, there might be a printing error? proportion of infants > 5 years, should be < 5 years?

Thank you. We reviewed it and verified the 2017 national census for update the information:

In line 194:

IT SAYS: It has the lowest proportion nationally of children >5 years old (5.1%).

IT SHOULD SAY: It has the lowest proportion nationally of children <5 years old (7,3%).

INEI. Censos Nacionales 2017 URL: < <http://censo2017.inei.gob.pe/> >

1. Do the authors know the proportion of stunting for region 1, as they have described for region 2 and 3?

The following is now included: Stunting for Lima in 2019 was reported as 4.9%. INEI. DHS 2019. URL<<https://proyectos.inei.gob.pe/endes/ppr.asp>>

2. Could they also provide the proportion of infants < 5 years for region 2 and 3?

The following is now included: Ayacucho 8,8% and Ucayali 11.8%. INEI. Censos Nacionales 2017 URL: < <http://censo2017.inei.gob.pe/> >

Then the description of the recruitment populations from the 3 regions would be cohesive. If any of this data is not known that could be stated in the description.

VERSION 2 – REVIEW

REVIEWER	Heike Rabe Brighton and Sussex Medical School, UK
REVIEW RETURNED	12-Apr-2021

GENERAL COMMENTS	My comments have been addressed, thank you.
---

REVIEWER	Ann Von Holle National Institute of Environmental Health Sciences, United States
REVIEW RETURNED	15-Apr-2021

GENERAL COMMENTS	I appreciate the revisions that address the comments based on the initial submission.
---